# *Escherichia coli* Isolated from Organic Laying Hens Reveal a High Level of Antimicrobial Resistance despite No Antimicrobial Treatments

**DOI:** 10.3390/antibiotics11040467

**Published:** 2022-03-30

**Authors:** Claudia Hess, Salome Troxler, Delfina Jandreski-Cvetkovic, Angelika Zloch, Michael Hess

**Affiliations:** 1Clinic for Poultry and Fish Medicine, Department for Farm Animals and Veterinary Public Health, University of Veterinary Medicine, Veterinaerplatz 1, 1210 Vienna, Austria; salome.troxler@vetmeduni.ac.at (S.T.); delfina.jandreski@vetmeduni.ac.at (D.J.-C.); zloch@mitsch.co.at (A.Z.); michael.hess@vetmeduni.ac.at (M.H.); 2Tierarzt GmbH Dr. Mitsch, Hauffgasse 24, 1110 Vienna, Austria

**Keywords:** antibiotics, susceptibility, resistant, poultry, organic laying hens, *E. coli*, microdilution, MDR, XDR, PDR

## Abstract

The present study investigated the resistance characteristics of *E. coli* isolates originating from 18 organic laying hen flocks. *E. coli* was isolated from different organs at three different time points, resulting in 209 *E. coli* isolates. The antibiotic susceptibility was determined by applying a microdilution assay. General, a high resistance rate was found. The antibiotic susceptibility was independent from the presence of pathological lesions, the isolation site, or the affiliation to a pathogenic serogroup. The majority of the isolates proved to be multi-drug-resistant (95.70%), of which 36.84% could be categorized as extensively drug-resistant. All isolates were resistant to oxacillin and tylosin. Resistance rates to amoxicillin (67.94%), cefoxitin (55.98%), ceftazidime (82.30%), colistin (73.68%), nalidixic acid (91.87%), streptomycin (42.58%), tetracycline (53.59%), and sulfamethoxazole (95.22%) were high. None of the isolates revealed pan-drug-resistance. A great heterogeneity of resistance profiles was found between isolates within a flock or from different organs of the same bird, even when isolates originated from the same organ. An increase in antimicrobial resistance was found to be correlated with the age of the birds. The fact, that no antibiotic treatment was applied except in two flocks, indicates that resistant bacteria circulating in the environment pose a threat to organic systems.

## 1. Introduction

Antimicrobial resistance (AMR) is one of the major threats to animal and human health. As a result, it is considered an important issue within the One Health approach, as it also affects food safety. Worldwide, antimicrobials are used in food-producing animals to combat bacterial diseases and ensure productivity [1]. Previous studies have shown that the inappropriate use of such substances may lead to an increase in resistant bacteria [2,3,4]. One intervention strategy to counter this scenario is the limitation of antimicrobial usage in animal production, as implemented for organic production based on the Commission Regulation (EC) No. 889/2008 [5]. This regulation dictates that a maximum of three antimicrobial treatments are allowed during a production period of 12 months, and if the production period is shorter than 12 months, the application of antibiotics is only allowed once. Additionally, the withdrawal period after the use of antibiotics has to be doubled. In EU agriculture, alternative husbandry systems for laying hens are a fast-growing area and are already used for more than half of these birds, including 6.2% of birds in organic systems [6].

*Escherichia* (*E.*) *coli* is a commensal bacteria in the gut of chickens, but they can also cause severe diseases in poultry [7]. Furthermore, there is increasing evidence that these bacteria can be transmitted via the food chain to humans [8,9,10,11,12]. The appearance of antibiotic resistance in *E. coli* is seen as an indicator for transmission of resistance within bacterial populations being a benefit for the spread of resistant bacteria [13,14]. In Austria, 12.5% of laying hens are kept in organic farming systems that implement the abovementioned legislation, with restrictions on applying antimicrobial substances. Therefore, it can be speculated that *E. coli* isolates from such birds carry less antibiotic resistance. In the present investigation, 18 organic laying flocks were sampled three times during their production—before the start of lay, at the peak of lay, and at the end of lay—and *E. coli* was isolated from different organs. By applying an antibiotic susceptibility test using the microbroth dilution method the overall antibiotic susceptibility was determined in the first step. Furthermore, differences in the antibiotic resistance profiles of isolates within a flock, between isolates of different organs of the same bird, and between isolates from one organ were determined. Finally, the influence of a bird’s age on the antibiotic resistance of *E. coli* isolates was investigated.

## 2. Results

Birds from two flocks received antibiotic treatment once during their lifetime: flock 1 received tylosin at the age of 10 weeks, and flock 12 received colistin at the age of 26 weeks. Except for flock 12, none of the farmers reported outbreaks of colibacillosis. No pathological lesions were found in birds necropsied at the first sampling time point (S1, pullets). At the second (S2, peak of lay) and third sampling time points (S3, end of production) one up to 4 birds each were affected by oophoritis and salpingitis in 4 and 10 flocks, respectively (Table 1).

In total, 209 *E. coli* field isolates were obtained and investigated. At the first time point, 47 isolates were derived; at the second, 70 isolates; and at third, 92 isolates were obtained. Only 34 isolates could be attributed to a specific serogroup: 29 isolates were proven to be O1, and 5 were proven to be O2. The majority of *E. coli* isolates originated from the reproductive tract (S1 n = 34; S2 n = 59; S3 n = 73) (Table 2).

All isolates were resistant to oxacillin and tylosin (Table 3). Furthermore, within the antimicrobial class of penicillins, reduced susceptibility was found for amoxicillin, with 67.94% of isolates proving resistant. High resistance rates were also detected in the case of two cephalosporins, cefoxitin (55.98%) and ceftazidime (82.30%), as well as for colistin (73.68%), nalidixic acid (91.87%), streptomycin (42.58%), tetracycline (53.59%), and sulfamethoxazole (95.22%). A low number of isolates showed resistance to gentamicin (6.22%) and neomycin (11.48%).

Interestingly, multidrug resistance (MDR) was found in the majority of isolates (200/209). Seventy-seven isolates proved extensively drug-resistant (XDR), but none of the isolates could be attributed to the category of pan-drug-resistant (Figure 1). A great heterogeneity of antimicrobial resistance profiles was found not only between isolates within a flock, but also between isolates from different organs of the same bird, and even between isolates originating from the same organ (Table 4).

No influence on the antimicrobial resistance was found based on the isolation sites of *E. coli*, as the proportion of resistance was 45.39% for group 1 (heart, liver, lungs) and 43.93% for group 2 (ovary, oviduct) (Figure 2). Interestingly, an increase in resistant isolates was found as the age of the birds increased. The mean proportion of antimicrobials to which the *E. coli* isolates were resistant was 39.92% (8.38/21) at S1, compared to S2 and S3, with 47.80% (n = 10.04/21) and 47.00% (9.87/21), respectively (Figure 3).

## 3. Discussion

Worldwide, the increase in antibiotic resistance has raised general concern, and factors such as diseases outbreaks and antibiotic interventions could drive the emergence of AMR bacteria in food-producing animals by inappropriate use of antimicrobials [2]. Consequently, previous studies have emphasized the importance of reducing antimicrobial usage in animals [15,16,17,18]. In this regard, organic farming is an interesting area, as it differs from conventional production in the strict limitations put in place for antimicrobial application. To monitor the progress of any intervention, continuous surveillance programs are necessary. For this, indicator bacteria, such as *E. coli*, commonly found in healthy animals and known to acquire AMR faster than other bacteria, are generally used [19,20]. Previous studies revealed that, although *E. coli* isolates from organic laying hens showed higher susceptibility rates compared with isolates from conventional housing systems, a notable number of resistant isolates were still present [21,22,23]. One hypothesis for this finding was that more time might be needed to increase the susceptible bacterial population, even with the lack of antibiotics, as not all types of resistance are maintained for an equally long time period in a given environment after removal of the selecting antimicrobial substance [24]. Among the EU member states, Austria was the first to ban conventional cages in 2009, and about 12.5% of layers are kept in organic husbandry systems [25]. This is an excellent position to gain actual data on the antimicrobial susceptibility of *E. coli*. The present study monitored 18 organic laying hen flocks from their pullet age until the end of their production life, and *E. coli* was isolated from different organs. Their antibiotic susceptibility was independent of the presence of typical pathological lesions, indicative of colibacillosis, and the site of isolation, a finding in agreement with recently reported data [26,27]. Furthermore, only 16.75% of isolates could be attributed to the pathogenic serogroups O1 and O2, highlighting the copresence of heterogenic *E. coli,* as has been reported previously [28]. Only two flocks were treated with antibiotics once during their life. Therefore, the relationship between the application of antibiotics and the susceptibility of *E. coli* could not be assessed in the present study.

The general finding that all *E. coli* isolates were resistant to oxacillin and tylosin is not surprising, as natural resistance is a well-known phenomenon [29]. Interestingly, higher resistance rates were found for penicillins, namely ampicillin and amoxicillin, which is more in agreement with data reported from conventional flocks [19,21,30,31]. Additionally, only a low number of isolates proved to be susceptible to some substances from the antimicrobial classes of the second- and third-generation cephalosporins. In this context, it was revealed that the use of ceftiofur in layer hatcheries contributed to an increase in the number of resistant *E. coli* isolates [32,33]. However, in Austrian hatcheries, no antibiotic substances are used. Conflicting data are reported with regard to susceptibility to chloramphenicol and aminoglycosides, ranging from high numbers of susceptible isolates to a substantial number of resistant isolates [21,23,30,32]. The latter finding might be linked to plasmid-mediated resistance against cephalosporins as third-generation cephalosporin resistance genes and other antimicrobial-resistance genes are linked together on the plasmid [34]. Interestingly, the majority of isolates proved susceptible to neomycin, which is widely used in Austria to treat colibacillosis in laying hens (personal communication with field veterinarians). Beside this antibiotic, colistin is most often applied. Here, the resistance rate proved surprisingly high (approximately three quarters of the isolates), although it was applied only once in one flock. It is known that colistin resistance can rapidly spread between bacteria via the transferable plasmid-mediated *mcr-1* gene resulting in a stable resistant bacterial population in certain geographic areas [35,36,37]. The present finding is in agreement with other studies that report a high prevalence of colistin-resistant *E. coli* in chickens [26,37,38,39], but is in clear contrast to previous data that report nearly exclusively susceptibility [21,23,31]. This also has a special implication for human medicine, as the WHO included colistin in their Essential Medicines List Access, Watch, and Reserve (AWaRE) classification [40].

For quinolones, the number of *E. coli* isolates susceptible to enrofloxacin was found to be much higher than for nalidixic acid, a fact also known from healthy broiler chickens [41]. This aspect can be attributed to the ability of chromosomal mutations in DNA gyrase and topoisomerase IV as well to horizontal spread via plasmids [42,43,44]. Therefore, the reduction of a homologous substance within an antimicrobial class may not result in the minimization of the resistance to a specific antibiotic [23]. The presence of high numbers of tetracycline-resistant *E. coli* isolates in chickens is well-known and in agreement with the present results [19,21,22,23]. Beside a linkage to plasmid-mediated cephalosporin resistance, tetracyclines are known to persist in animal manure and may potentially lead to the persistence of antibiotic-resistant bacteria in the environment [33,45]. A general tendency towards increasing numbers of sulfonamide-resistant *E. coli* isolates was reported by different groups [19,23,46]. In the present investigation, hardly any isolates were found to be susceptible to sulfamethoxazole, but the majority were still found to be susceptible to the combination of trimethoprim and sulfamethoxazole. Sulfonamide-resistance genes are known to be easily transferred between commensal *E. coli* via integrons, transposons, or plasmids [47]. As a result, a constant sulfonamide-resistant bacterial community might be present.

Keeping in mind that only two of the flocks experienced an antibiotic treatment during their whole lifetime, the high prevalence of resistant isolates as well as the fact that the vast majority of them could be attributed to the category of MDR (approximately one third of which could be classed as XDR), is a remarkable finding. Furthermore, the present study revealed an increase in AMR isolates in relation to the increase in the birds’ age. So far, this result has either been reported for young chickens or found to be independent of the age in laying hens [38,48]. Housing conditions with good management practices were reported to influence the prevalence of resistant bacteria, as access to pasture was related to higher numbers of resistant isolates [24,49]. Recently, it was demonstrated that the presence of the same cephalosporin-resistant *E. coli* isolate could be found in broilers from two consecutive flocks and in the broiler house environment [50]. Consequently, the transmission of antibiotic-resistant *E. coli* may occur through fecally contaminated environments/manure, serving as an intermediate habitat where striking changes in *E. coli* populations and their antibiotic resistance patterns, as well as the accumulation of resistant isolates and their dissemination, were reported [8,51,52,53,54]. Furthermore, it is also known that the gut commensals take up resistance genes, thereby serving as a reservoir to transmit resistance [55,56,57]. These facts might explain the present findings.

In this study, high resistance was found in *E. coli* isolates derived from organic laying flocks, proving MDR despite the lack of antimicrobial treatments. This finding was associated with great heterogeneity in the susceptibility patterns, not only within a flock but also within a bird, and even within an organ. We were also able to show that the number of resistant isolates increased with the age of the birds. The presented data lead to the assumption that the environment might play an important role being a central habitat for the accumulation of antibiotic resistant bacteria in organic farming systems.

## 4. Materials and Methods

### 4.1. Study Design

The study design and flock data were previously reported in [58]. Briefly, on a voluntary basis, 18 farmers of organic laying hen farms, with the consent of their field veterinarians, participated in the present study. Flocks no. 13 and 14 were located at the same farm. The remaining 16 flocks originated from one farm each. The flock size ranged from 1270 to 6000 birds (average: 4080 birds). Each flock was sampled at three time points: before onset of lay (S1—age 14 to 19 weeks, mean 17 weeks); at peak of lay (S2—age 35 to 40 weeks, mean 37 weeks); and at end of lay (S3—age 63 to 79 weeks, mean 72 weeks). Samples before onset of lay were taken on the rearing farm before transfer of birds to the laying farms. If multiple laying flocks were supplied by the same pullet flock, only one sampling was performed, resulting in a total of 13 sampled pullet flocks. At each sampling time point, five freshly dead birds per flock were collected. For each bird, the occurrence of gross pathological lesions with a focus on the reproductive tract lesions, namely oophoritis and salpingitis, were recorded.

### 4.2. Bacteriology

The following organs were sampled by cutting the surfaces aseptically: ovary, oviduct, heart, liver, and lung. After cutting the surfaces, material was taken from the inner organ parts and inoculated directly on MacConkey Agar (LABM, Lancashire, United Kingdom) by performing the streak plate procedure using sterile inoculation loops (10 µL, VWR, Vienna, Austria). The agar plates were incubated aerobically for 24 h at 37 °C. *E. coli* isolates were identified by their growth on MacConkey Agar within 24 h characterized by circular, convex, smooth, and bright pink colonies with an entire margin and surrounded by a precipitate [59]. By applying the direct transfer method following the manufacturers’ protocol MALDI-TOF MS (Bruker Daltonics GmbH, Bremen, Germany) confirmed the isolates as *E. coli* as reported previously [58]. The determination of serogroup O1, O2, and O78 was performed by slide agglutination test (SIFIN diagnostics GmbH, Berlin, Germany), as reported previously [28].

### 4.3. Antimicrobial Susceptibility Test

The susceptibility of *E. coli* isolates was determined according to CLSI supplement VET06 [60], CLSI supplement M100 [61], CLSI standard VET01 [62], and CASFM recommendations [63] by using an individually designed MICRONAUT-S Veterinary plate (MERLIN Diagnostika GmbH, Bornheim-Hersel, Germany). The antimicrobial substances and their concentrations are given in Table 1. The preparation of the bacterial test suspensions was carried out according to the manufacturers’ instructions. Briefly, a bacterial suspension in 5 mL NaCl, according to the McFarland 0.5 standard, was made. From this, 50 µL was transferred into 11 mL Mueller–Hinton broth (Merck, Vienna, Austria) and mixed well. Each well was filled with 100 µL of this suspension. All inoculated microtiter plates were incubated aerobically for 24 h at 37 °C. The evaluation of results was performed using MCN6 software, version 6.00, release 72 (MERLIN Diagnostika GmbH, Bornheim-Hersel, Germany).

### 4.4. Antimicrobial Resistance Profiles

Drug resistance was categorized based on Magiorakos et al. [64]. Briefly, MDR was defined as acquired nonsusceptibility to at least one substance in three or more antimicrobial classes; XDR as nonsusceptibility to at least one substance in all but two or fewer antimicrobial classes; and pan-drug-resistant (PDR) as nonsusceptibility to all substances in all antimicrobial classes. For this analysis, results from oxacillin and tylosin were excluded.

### 4.5. Analyses of Data

Descriptive analyses were performed to detect influences of the isolation site and the age of the birds on the antimicrobial susceptibility of *E. coli.* For each isolate, the number of resistance (number of antimicrobials to which the isolate was resistant) was determined. To describe the influence of isolation sites, isolates were attributed to two groups: group 1 (comprising isolates from heart, liver, and lungs) and group 2 (isolates from ovaries and oviducts). The grouping of isolates according to the sampling time point was carried out to investigate the influence of the birds’ age. Data were calculated and visualized using Microsoft Excel 2016. The categorization into MDR, XDR, and PDR classes was performed according to Magiorakos et al. [62], by applying the criteria described above.

## Figures and Tables

**Figure 1 antibiotics-11-00467-f001:**
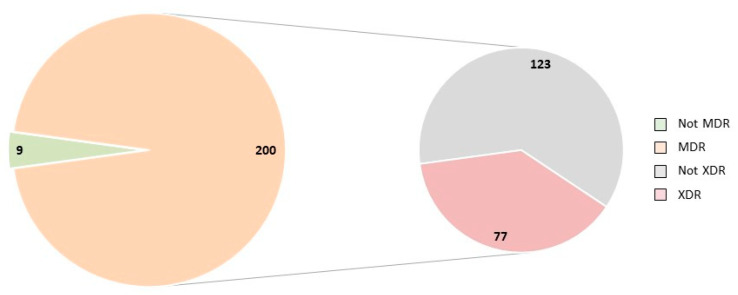
Categorization of the 209 *E. coli* isolates investigated based on their antimicrobial resistance profiles into multidrug resistant (MDR) and non-MDR, and their further subdivision into extensively drug-resistant (XDR) and non-XDR Not MDR—isolates without multidrug resistance; MDR—isolates with multidrug resistance; not XDR—isolates without extensive drug resistance; XDR—isolates with extensive drug resistance.

**Figure 2 antibiotics-11-00467-f002:**
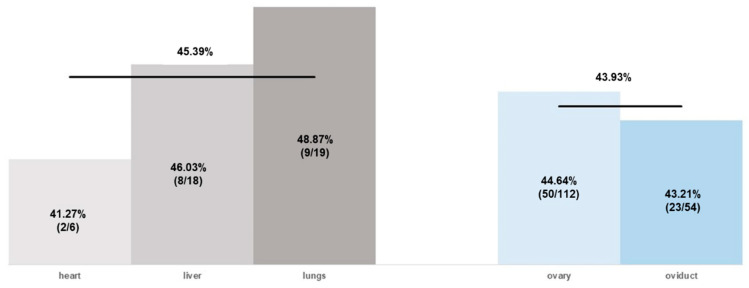
Percentage proportion of *E. coli* resistance presented as mean value by comparing the isolates based on their isolation site (total numbers are given in brackets). Isolates from the heart, liver, and lungs were allocated to group 1, and isolates from ovaries and oviducts to group 2.

**Figure 3 antibiotics-11-00467-f003:**
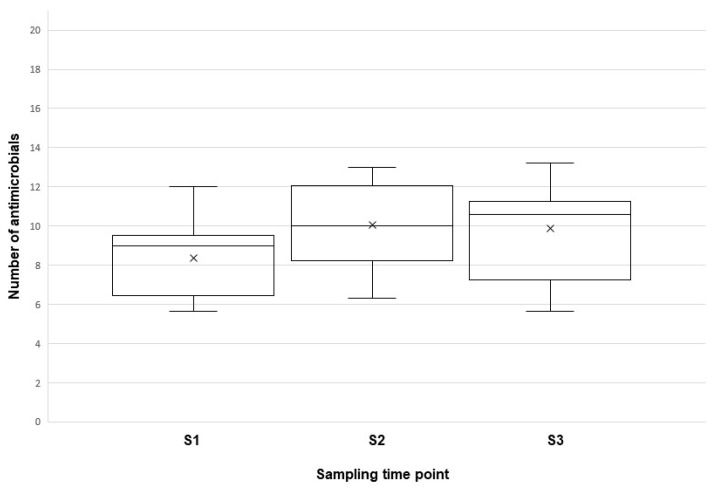
Number of antimicrobial substances to which *E. coli* isolates were resistant, shown in relation to the age of birds (S1 = sampling point 1, age ~17 weeks; S2 = sampling point 2, age ~38 weeks; S3 = sampling point 3, age ~74 weeks). The mean number of antimicrobials to which *E. coli* isolates were resistant was 8.38 for S1, 10.04 for S2, and 9.87 for S3.

**Table 1 antibiotics-11-00467-t001:** Occurrence of pathological lesions in ovary/oviduct.

Flock No.	Pathological Lesions in Ovary/Oviduct		
	S1	S2	S3
1	0/5 ^b^	2/5	0/5
2	0/5	0/5	0/5
3	0/5	0/5	2/5
4	0/5	1/5	4/5
5	0/5	0/5	0/5
6	0/5	0/5	0/5
7	0/5	0/5	1/5
8	0/5	0/5	0/5
9	0/5	0/5	1/5
10	0/5	0/5	2/5
11	0/5	0/5	0/5
12	0/5	0/5	4/5
13 ^a^	0/5	0/5	0/5
14 ^a^	0/5	2/5	1/5
15	0/5	0/5	2/5
16	0/5	0/5	2/5
17	0/5	0/5	4/5
18	0/5	0/5	0/5

^a^ flock no. 13 and 14 originated from the same farm; ^b^ number of birds with pathological lesions in ovary and oviduct/number of necropsied birds.

**Table 2 antibiotics-11-00467-t002:** Number of *E. coli* isolates per organ isolated at S1, S2, and S3, with corresponding serogroup in brackets.

Flock	Time Point														
	1					2					3				
	Heart	Liver	Lung	Ovary	Oviduct	Heart	Liver	Lung	Ovary	Oviduct	Heart	Liver	Lung	Ovary	Oviduct
1					5 (5 × O1)	1		1	1 (O2)	1		1		4	
2				4					4 (1 × O1)			1 (O1)			4 (3 × O1)
3					1										4
4	1			1	1			2		3				9	6
5		2	2	6					4 (1 × O1)						3
6			2										3	3 (1 × O1)	
7	2			2						3 (1 × O1)					
8			3	3	3				6	3					
9			1	3 (1 × O1)			1 (O1)	1	1					5 (2 × O1)	2
10									5 (4 × O1)			2 (1 × O1)		2 (2 × O1)	
11									5 (4 × O1)				2	1	
12				4 (1 × O1)	1				4	1		1	2	1	1 (O2)
13									2 (1 × O2)	2		1		3	1
14									3 (2 × O2)	6		3		3	
15						2	3		5						
16														3	3
17														15	
18												3			
Total	3	2	8	23	11	3	4	4	40	19	0	12	7	49	24

**Table 3 antibiotics-11-00467-t003:** Antimicrobial substances and concentrations used for AMR testing, respective minimal inhibitory concentration for resistance (MIC, µg/mL), and the percentage of resistant isolates based on the given MIC.

Class	Antimicrobial Substance	Concentrations (µg/mL)										MIC (µg/mL)	% of Resistant Isolates (n = 209)
Penicillin, penicillin combination	Amoxicillin	4	8	16	32							≥32	67.94%
	Amoxicillin/clavulanate	4/2	8/4	16/8	32/16							≥32/16	2.39%
	Ampicillin	0.25	0.5	1	2	4	8	16				>16	17.70%
	Oxacillin	0.25										>0.2	100.00%
Cephalosporin	Cefazolin (1st generation)	2	4									>4	20.10%
	Cefoxitin (2nd generation)	4										>4	55.98%
	Cefotaxim (3rd generation)	0.25	0.5	1	2	4	8	16	32			≥4	9.57%
	Ceftazidim (3rd generation)	0.25	0.5	1	2	4	8	16	32			≥16	82.30%
Chloramphenicol	Chloramphenicol	4	8	16	32							≥32	33.97%
Polypeptide	Colistin	0.0313	0.063	0.13	0.25	0.5	1	2	4	8	16	≥4	73.68%
Quinolone	Enrofloxacin	0.125	0.25	0.5	1	2						≥2	11.96%
	Nalidixic acid	4	8	16	32	64						>64	91.87%
Aminoglycoside	Gentamicin	1	2	4	8	16						≥8	6.22%
	Neomycin	4	8	16	32							≥16	11.48%
	Streptomycin	8	16	32	64							≥32	42.58%
Carbapenem	Imipenem	1	2	4								≥4	9.09%
Tetracycline	Tetracycline	0.25	0.5	1	2	4	8	16				≥16	53.59%
Diaminopyrimidine, sulfamethoxazole, and combinations	Trimethoprim	8	16									≥16	37.80%
	Sulfamethoxazole	256	512									≥512	95.22%
	Trimethoprim/sulfamethoxazole	0.5/9.5	1/19	2/38	4/76							≥4/76	14.83%
Macrolide	Tylosin	1	2	4	8	16						≥16	100.00%

**Table 4 antibiotics-11-00467-t004:** Differences in antimicrobial profiles of *E. coli* isolates within a flock, between organs of the same bird, and within the same organ, exemplarily shown for flock 6 (excluding oxacillin and tylosin). Results of susceptibility testing are given as susceptible (S), resistant (R), and intermediate (I).

Time Point	Organ	Antimicrobials																		
		AMC	AMP	AMX	CAZ	CEZ	CMP	COL	COX	CTX	ENR	GEN	IMP	NAL	NEO	SMO	STR	T/S	TET	TRP
1	oviduct	S	R	R	R	I	S	S	R	S	I	S	R	R	S	R	I	S	R	S
1	oviduct	S	S	S	R	I	S	R	S	S	I	S	S	R	S	S	R	S	S	S
1	oviduct	S	S	S	R	S	S	R	S	S	I	S	S	R	S	R	I	S	S	S
1	oviduct	S	S	S	S	I	S	S	R	S	R	R	R	R	R	R	R	S	I	S
1	oviduct	S	S	S	S	I	S	S	S	S	S	S	S	S	S	R	S	S	S	S
2	heart	S	S	R	R	S	S	R	R	S	I	S	S	R	S	R	S	R	R	R
2	lung	S	S	R	R	S	I	R	R	S	I	S	S	R	I	R	I	R	R	R
2	ovary	S	I	R	R	S	R	R	R	S	I	S	S	R	I	R	R	R	R	R
2	ovary	S	S	I	R	S	R	R	S	S	I	S	S	R	I	R	S	R	R	R
2	ovary	S	S	R	R	S	I	R	R	S	I	S	S	R	S	R	I	S	R	R
2	oviduct	S	S	S	R	S	S	R	R	R	I	I	R	R	S	R	R	R	R	R
3	liver	S	S	R	R	S	R	R	S	S	R	S	S	R	R	R	R	S	R	R
3	ovary	R	R	R	R	R	R	R	S	R	R	R	S	R	R	R	R	R	R	R
3	ovary	S	S	R	R	S	R	R	R	S	I	I	S	R	I	R	I	S	R	R
3	ovary	S	S	R	R	S	R	R	S	I	I	S	S	R	I	R	R	S	R	R

AMC—amoxicillin; APM—ampicillin; AMX—amoxicillin/clavulanate; CAZ—ceftazidim; CEZ—cefazolin; CMP—chloramphenicol; COL—colistin; COX—cefoxitin; CTX—cefotaxim; ENR—enrofloxacin; GEN—gentamicin; IMP—imipenem; NAL—nalidixic acid; NEO—neomycin; SMO—sulfamethoxazole; STR—streptomycin; T/S—trimethoprim/sulfamethozaxole; TET—tetracycline; TRP—trimethoprim.

## Data Availability

Not applicable.

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
