# Peer review of "Escherichia coli Isolated from Organic Laying Hens Reveal a High Level of Antimicrobial Resistance despite No Antimicrobial Treatments"

_antibiotics, 2022, doi:10.3390/antibiotics11040467_

Round 1

Reviewer 1 Report

Antibiotics, 2022, entitled „Escherichia coli isolated from organic laying hens reveal a high level of microbiological resistance despite no antimicrobial treatments”  

E. coli isolates originating from several organic layer hen flocks were isolated from various organs at three different time points and in total 209 E. coli isolates were characterized. The resistance profiles to various antimicrobial substances were heterogeneous between isolates within a flock, between organs of an animal and within one organ. Interestingly, high percentages of antimicrobial resistances were identified, even if no antibiotic was used as a feed additive. Most of the isolates showed multi-drug resistance. The main message indicated that resistant microorganisms circulating in the environment. An important statement is the increase of antimicrobial resistance in dependence on the age of the hen. The older the animal, the more stable the bacteria to antimicrobials. The authors found out that 200 of the 209 isolates demonstrate a MDR phenotype, of which 123 were MDR and 77 were XDR.

The results in the manuscript are presented clearly and the manuscript is well written. The data and statistics are well presented.

The authors assessed the phenotypes of the isolates in terms of antimicrobial resistances. The question arises about the origin of the resistances, perhaps by food or water. However, the question arises whether these are actually resistances or whether these are tolerances. Genetic elements and resistance genes can originate from the environment and be taken up by E. coli. In this case, the phenotype of a resistance due to the new genotype remains. If these are tolerances, the stable antimicrobial phenotype should be cancel out. Therefore, it is difficult to differentiate whether these are tolerances or resistances. A basis of a resistance may lay on the molecular level with the detection of the resistance genes. A tolerance come up during bacterial life, mainly under stress conditions. Aging supports the emergence of a tolerance on the one hand. On the other hand, elder bacteria that live longer in the hens are likely to pick up resistance genes from the environment more easily or be exposed to the genes for longer periods of time. Therefore, the description of a resistance is to be regarded as critical without evidences. As E. coli were isolated on MacConkey agar plates which are not selective to antimicrobial substances, it would be of interest to look to growth on selective plates as ESBL plates adapted plates comprising antimicrobial substances. Growth indicated actual resistance. And resistance can also be detected at the molecular level with PCR. This differentiation would be important for the statement whether the bacteria are taken up from the environment as genotypically resistant bacteria and become established in the animals or whether the bacteria develop phenotypically to the tolerances to antimicrobial substances.    

Here are some detail remarks:

Line 15: “isolates” are mentioned as “strains”. However, strains are defined as reference strains, which are sequenced or well known. In this context, “isolates” are meant. Please change the word in the whole text.

Line 119: Reference 24, without names

Line 134: References in brackets

Author Response

We thank the reviewer for the critical comments and suggestions which helped to enhance the quality of the manuscript.

Reviewer 1:

Antibiotics, 2022, entitled „Escherichia coli isolated from organic laying hens reveal a high level of microbiological resistance despite no antimicrobial treatments”  

  1. coli isolates originating from several organic layer hen flocks were isolated from various organs at three different time points and in total 209 E. coli isolates were characterized. The resistance profiles to various antimicrobial substances were heterogeneous between isolates within a flock, between organs of an animal and within one organ. Interestingly, high percentages of antimicrobial resistances were identified, even if no antibiotic was used as a feed additive. Most of the isolates showed multi-drug resistance. The main message indicated that resistant microorganisms circulating in the environment. An important statement is the increase of antimicrobial resistance in dependence on the age of the hen. The older the animal, the more stable the bacteria to antimicrobials. The authors found out that 200 of the 209 isolates demonstrate a MDR phenotype, of which 123 were MDR and 77 were XDR.

The results in the manuscript are presented clearly and the manuscript is well written. The data and statistics are well presented.

The authors assessed the phenotypes of the isolates in terms of antimicrobial resistances. The question arises about the origin of the resistances, perhaps by food or water. However, the question arises whether these are actually resistances or whether these are tolerances. Genetic elements and resistance genes can originate from the environment and be taken up by E. coli. In this case, the phenotype of a resistance due to the new genotype remains. If these are tolerances, the stable antimicrobial phenotype should be cancel out. Therefore, it is difficult to differentiate whether these are tolerances or resistances. A basis of a resistance may lay on the molecular level with the detection of the resistance genes. A tolerance come up during bacterial life, mainly under stress conditions. Aging supports the emergence of a tolerance on the one hand. On the other hand, elder bacteria that live longer in the hens are likely to pick up resistance genes from the environment more easily or be exposed to the genes for longer periods of time. Therefore, the description of a resistance is to be regarded as critical without evidences. As E. coli were isolated on MacConkey agar plates which are not selective to antimicrobial substances, it would be of interest to look to growth on selective plates as ESBL plates adapted plates comprising antimicrobial substances. Growth indicated actual resistance. And resistance can also be detected at the molecular level with PCR. This differentiation would be important for the statement whether the bacteria are taken up from the environment as genotypically resistant bacteria and become established in the animals or whether the bacteria develop phenotypically to the tolerances to antimicrobial substances.   

We thank the reviewer for this very valid comment. The focus of the present study was to generate results on the general antibiotic resistance as well as on differences in antimicrobial profiles based on isolation sites and age of birds of E. coli strains derived from layer hen flocks which are hardly treated with antimicrobials. Our intention was to provide actual practical data for the field. For this, we applied a microdilution assay and report AMR phenotypes which are of high relevance to the field. Therefore, in the discussion we addressed different facts which are known to contribute to antibiotic resistance in the absence of application/treatments (e.g. line 148 time-dependent increase of susceptible bacteria, line 167 use of antibiotics in hatcheries, lines 176, 184, 191 plasmid-mediated resistances, line 188 persistence of antibiotic-resistance bacteria in the environment). These facts are summarized in the lines 196-220 highlighting the indication for the environment in combination with the gut commensals to serve as reservoir for antibiotic resistant.

Screening for ESBL was not in the focus of the present study. But, of course we were surprised that the majority of strains could be phenotypically attributed to ESBL. This finding needs further investigation in which the question regarding tolerance/resistance will be addressed.

Here are some detail remarks:

Line 15: “isolates” are mentioned as “strains”. However, strains are defined as reference strains, which are sequenced or well known. In this context, “isolates” are meant. Please change the word in the whole text.

We thank the reviewer for this comment, and revised the manuscript accordingly.

Line 119: Reference 24, without names

Line 150: Revised accordingly.

Line 134: References in brackets

Line 165: Revised accordingly.

Reviewer 2 Report

Hess et al investigated the prevalence of antibiotic resistant E coli in the poultry farms. The study reveals interesting results that is important to the antimicrobial stewardship. It should be commented that, they have have tested different organs for E coli. However, there are some concerns that should be addressed before publishing this manuscript. First of all, the English language should be improved. Secondly, all the figure legends are short and you include little more details such as expansion of acronyms. Especially for figures 3 and 4, you can explain the median. Also include the numbers (nominator and denominator) where the percentage is given. 

Materials and methods: Break into smaller sections. Antimicrobial susceptibility test can be one section itself. Need reference for some of the material section such as bacterial culturing.

Statistical analysis: This is section has to be modified heavily. Need to explain what kind of statistical test is used for each result.

Overall, there is enough data to publish, but, need thorough revision.

Author Response

We thank the reviewer for the critical comments and suggestions which helped to enhance the quality of the manuscript.

Reviewer 2:

Hess et al investigated the prevalence of antibiotic resistant E coli in the poultry farms. The study reveals interesting results that is important to the antimicrobial stewardship. It should be commented that, they have have tested different organs for E coli. However, there are some concerns that should be addressed before publishing this manuscript.

First of all, the English language should be improved.

The English was reviewed accordingly.

Secondly, all the figure legends are short and you include little more details such as expansion of acronyms. Especially for figures 3 and 4, you can explain the median. Also include the numbers (nominator and denominator) where the percentage is given. 

We revised the figures and figure legends accordingly.

Materials and methods: Break into smaller sections. Antimicrobial susceptibility test can be one section itself. Need reference for some of the material section such as bacterial culturing.

We thank the reviewer for this helpful comment. Lines 214-273: Materials and methods were adapted accordingly. References were included.

Statistical analysis: This is section has to be modified heavily. Need to explain what kind of statistical test is used for each result.

This comment is in contrast to the opinion of the two other reviewers who judged the presentation of data/statistics as being very well. Therefore, we tried to adjust the data analysis to hopefully meet the requirements of all three reviewers. Lines 257-273: We changed the heading to “Analysis of data”. The chapter was revised to make the descriptive analyses of the data better understandable.

Overall, there is enough data to publish, but, need thorough revision.

Reviewer 3 Report

In this manuscript, the authors reported the antimicrobial resistance profiles of E. coli isolated from hens with no prior exposure to antibiotics. Using microdilution assay, they found that the majority of isolated strains show resistance to more than one drug. Such a phenomenon seems to be related to the age of birds. The work was well-designed and generated results with high importance to the field. However, a few points need to be addressed to increase the soundness of the claims made in this paper and improve the overall readability.

  1. Line 60: Please define S1, S2, and S3 here if this is their first appearance in the article.
  2. Line 61. This sentence is not very easy to follow. Please consider making it two sentences.
  3. Table 3 and line 120: Is there a specific reason for using decimal commas here? Please consider unifying the sign usage throughout the manuscript.
  4. Line 83-87: Please provide the definitions for MDR and XDR if this is their first appearance in the article.
  5. Figure 1-3: Please provide figure captions.
  6. Figure 1 and 2: Some of the numbers are barely visible. Please adjust the color of the sectors (columns) or move the numbers out of the sectors (columns).
  7. Table 4: Please define S, R, and I.
  8. Figure 2: Please define the x-axis and y-axis. What is the reason behind zooming in on the y-axis? Any statistical analyses or evidence to support the claim in line 96?
  9. Line 97: Please elaborate because this sentence is a little unclear. Also, the authors should provide statistical analyses for any difference or trend observed.
  10. The Discussion section is rather long and wordy. The authors should consider making it more concise and centered around the findings from their experimental data.
  11. Please correct the inconsistency in in-text citation formatting.

Author Response

We thank the reviewer for the critical comments and suggestions which helped to enhance the quality of the manuscript.

Reviewer 3:

In this manuscript, the authors reported the antimicrobial resistance profiles of E. coli isolated from hens with no prior exposure to antibiotics. Using microdilution assay, they found that the majority of isolated strains show resistance to more than one drug. Such a phenomenon seems to be related to the age of birds. The work was well-designed and generated results with high importance to the field. However, a few points need to be addressed to increase the soundness of the claims made in this paper and improve the overall readability.

  1. Line 60: Please define S1, S2, and S3 here if this is their first appearance in the article.

Lines 60-62: Revised accordingly

  1. Line 61. This sentence is not very easy to follow. Please consider making it two sentences.

Lines 62-63: We thank the reviewer for this comment and recognized the problem. The sentence is revised accordingly.

  1. Table 3 and line 120: Is there a specific reason for using decimal commas here? Please consider unifying the sign usage throughout the manuscript.

We apologize for this mistake, and revised everything accordingly.

  1. Line 83-87: Please provide the definitions for MDR and XDR if this is their first appearance in the article.

We thank the reviewer for this comment. Lines 84-86: Revised accordingly.

  1. Figure 1-3: Please provide figure captions.

Figure captions were included in Figure 1. Figures 2 and 3 were changed according to the comments of this reviewer as well as of reviewer 2. Therefore, the figure legends were revised accordingly.

  1. Figure 1 and 2: Some of the numbers are barely visible. Please adjust the color of the sectors (columns) or move the numbers out of the sectors (columns).

We apologize for the previous bad quality of the figures. Both figures were revised by adjusting the colors, and including the comments from this reviewer as well as the comments from reviewer 2.

  1. Table 4: Please define S, R, and I.

We thank the reviewer for this comment, and apologize for this mistake. It is revised now accordingly.

  1. Figure 2: Please define the x-axis and y-axis. What is the reason behind zooming in on the y-axis? Any statistical analyses or evidence to support the claim in line 96?

We also thank for this comment. Figure 2 was revised based on the comments from this reviewer as well as the comments from reviewer 2.

Lines 108-110. We revised the sentence accordingly by including detailed data derived from analysis based on Figure 2.

  1. Line 97: Please elaborate because this sentence is a little unclear. Also, the authors should provide statistical analyses for any difference or trend observed.

Lines 110-113: The sentence was revised accordingly by including detailed data derived from analysis based on Figure 3.

  1. The Discussion section is rather long and wordy. The authors should consider making it more concise and centered around the findings from their experimental data.

The discussion was revised and shortened.

  1. Please correct the inconsistency in in-text citation formatting.

We apologize for this. The inconsistency was revised.

Round 2

Reviewer 1 Report

Thank you for the revisions. I have no comments.

Author Response

Dear Reviewer 1, Thank you very much.

Reviewer 2 Report

The authors have addressed the concerned raised in the initial review and now the manuscript is in acceptable form.

Author Response

Dear Reviewer 2, Thank you very much.